# Comparison of Transient Elastography and Point Shear Wave Elastography for Analysis of Liver Stiffness: A Prospective Study

**DOI:** 10.3390/diagnostics14060604

**Published:** 2024-03-12

**Authors:** Giuseppe Losurdo, Ilaria Ditonno, Domenico Novielli, Francesca Celiberto, Andrea Iannone, Antonino Castellaneta, Paola Dell’Aquila, Nunzio Ranaldo, Maria Rendina, Michele Barone, Enzo Ierardi, Alfredo Di Leo

**Affiliations:** 1Section of Gastroenterology, Department of Precision and Regenerative Medicine and Ionian Area, University of Bari, Piazza Giulio Cesare 11, 70124 Bari, Italy; ilaria.ditonno@gmail.com (I.D.); noviellidomenico@yahoo.it (D.N.); celibertofrancesca@gmail.com (F.C.); ianan@hotmail.it (A.I.); antocastellaneta@hotmail.com (A.C.); paoladellaquila@tiscali.it (P.D.); ranaldonunzio@alice.it (N.R.); mariarendina@virgilio.it (M.R.); michele.barone@uniba.it (M.B.); ierardi.enzo@gmail.com (E.I.); alfredo.dileo@uniba.it (A.D.L.); 2Ph.D. Course in Organs and Tissues Transplantation and Cellular Therapies, Department of Precision and Regenerative Medicine and Ionian Area, University of Bari, Piazza Giulio Cesare 11, 70124 Bari, Italy

**Keywords:** liver fibrosis, stiffness, fibroscan, shear wave elastography

## Abstract

Liver stiffness measurement (LSM) by Fibroscan is the most used non-invasive method to assess liver fibrosis. Recently, point-shear wave elastography (pSWE) has been introduced as a simple alternative non-invasive test. Therefore, we aimed to compare the results of these two techniques. One hundred and eighty-four consecutive patients attending our outpatient ultrasound clinic were recruited. LSM was performed by both Fibroscan and pSWE. Statistical analysis was conducted by Spearman’s test for correlation and linear regression. Bland–Altman graphs and ROC curves were drawn with area under the curve (AUC). Overall, the correlation of LS between Fibroscan and pSWE was substantial (r = 0.68, *p* < 0.001). Linear regression showed a coefficient b= 0.94 ± 0.02. The Bland–Altman plot found a bias of −0.10, with only 11 values exceeding the 95% confidence interval. When only considering patients with a LSM of > 10 kPa (*n* = 31), we found an excellent r = 0.79 (0.60–0.90, *p* < 0.001). A cutoff of 12.15 kPa for pSWE had sensitivity = 74.2% and specificity = 99.3% to detect relevant fibrosis, with an AUC = 0.98. The highest correlation was observed for hepatitis C (r = 0.91) and alcoholic liver disease (ALD)(r = 0.99). In conclusion, pSWE shows LSM estimation in agreement with Fibroscan in most cases, and the best concordance was observed for hepatitis C and ALD, and for higher ranges of LS.

## 1. Introduction

Chronic liver disease is a major public health issue and a major cause of death worldwide. In particular, non-alcoholic steatohepatitis (NASH) is becoming increasingly common, and it is expected to become the leading cause of liver transplantation [1]. The pathogenic basis of liver disease may vary from non-alcoholic fatty liver disease (NAFLD) to viral infections, autoimmune and genetic diseases, and alcohol abuse. Long-term lobular inflammation, both persistent or transient, regardless of the etiology, may eventually lead to liver fibrosis (deposition of collagen fibers associated with hepatocyte necrosis), while the remodeling of hepatic lobules may lead to cirrhosis, portal hypertension and, in most advanced cases, to hepatocellular carcinoma (HCC) [2]. Early diagnosis of liver fibrosis may influence patient management and the clinical decision-making process [3].

To date, the gold standard for diagnosing and evaluating liver fibrosis is liver biopsy with histological specimen examination by a pathologist. However, as with all invasive techniques, it has some limitations such as sampling errors and intra- and inter-observer variability mostly due to the limited sample size, not always representative of the whole liver parenchyma [4]. Complications such as potentially life-threatening bleeding and hemoperitoneum have been reported in the literature [5]; moreover, patients may reluctantly accept to undergo liver biopsy.

Hence, the scientific community’s interest in developing non-invasive techniques for liver fibrosis evaluation is compelling, not only at the moment of the diagnosis, but also over time due to the chronic nature of liver disease [6]. Most techniques focus on the assessment of liver stiffness as an indirect measurement of liver fibrosis with ultrasound-based elastographic methods. They differ in the physical properties used [7]. Another indirect way to measure hepatic fibrosis is the use of serum biomarkers, based on collagen metabolites detection in blood samples, although the specificity is very low, due to the interference of the eventual presence of inflammation outside the liver and modified clearance of collagen metabolites [2].

Transient elastography (TE), performed by Fibroscan, is based on shear waves that are generated by an external mechanical push through a vibrating device, generating a short duration vibration that spreads longitudinally compared to the transducer axis; meanwhile, the system measures the velocity of the wave [8].

Point shear-wave elastography (pSWE) is a more recent technique. It produces shear waves, which are transverse mechanical waves generated by the push pulse of a focused ultrasound beam through acoustic radiation force impulse [7,9]. The speed of shear waves is correlated with liver stiffness; indeed, shear waves travel faster in stiffer fibrotic tissue compared to healthy controls [10]. Point SWE differs from TE as it allows liver assessment through B-mode ultrasound image guidance, which helps in screening and diagnosing liver cirrhosis or focal liver lesions, and avoids both intrahepatic (e.g., bile ducts and blood vessels) and extrahepatic off-target measurements (kidneys, ribs, etc.), by selecting the correct acoustic window [11]. In pSWE, the measurement of stiffness is conducted by targeting a single point in the localized region of interest (ROI). This is different from 2D-SWE, the other shear wave elastography technique, in which the device calculates the mean stiffness within a larger area.

There have been several studies showing the accuracy of both techniques in assessing liver fibrosis [12], also when compared to liver biopsy, as it shows a linear relationship with the amount of fibrosis in hepatic tissue [11]. Therefore, our aim was to compare these two techniques for the non-invasive assessment of liver fibrosis, by evaluating their correlation and measure of diagnostic accuracy of pSWE compared to TE. These analyses were conducted in the overall population and according to the etiology of liver disease. 

## 2. Materials and Methods

This was a prospective single center study comparing liver stiffness measurements obtained by TE Fibroscan and pSWE. All examinations were carried out in the period November 2021–November 2022 by two experienced operators (GL and NR), with an expertise of more than 500 TE. The study was approved by the local Ethic Committee Board of Policlinico Universitary hospital of Bari (Prot. No. 0085420/06/10/2022).

Consecutive patients of either sex, without racial distinction and aged 18 years or older, attending the ultrasound facility of the outpatient Clinic of the Gastroenterology and Digestive Endoscopy Unit of Policlinico di Bari, were recruited. They underwent liver stiffness measurement by both TE via Fibroscan with M size probe (Echosens, Paris, France) and pSWE (QElaXto program on EsaoteMylab Xpro80, Genova, Italy), consecutively. Only patients capable of expressing informed consent for the study were recruited.

We excluded patients with central obesity, substantial thoracic fat, perihepatic ascites or acute hepatitis with hypertransaminasemia (Aspartate or alanine transaminase > ×5 upper limit of normality), right heart failure, very narrow intercostal spaces, or patients who had undergone liver transplantation or thoracic surgery with manipulation of the rib cage. Patients not eligible for liver stiffness measurement according to the physician’s opinion were also excluded from the study. 

During both examinations, patients were asked to maintain supine position and to hold their breath while the measurements are taken, with their arms maximally abducted; fasting was required on the day of the examinations [13]. At least 10 measurements had to be performed, positioning the probe in the regions of the upper right lobe of the liver. Kilopascal (kPa) was used as the unit of measurement for both techniques. Results were expressed as median with interquartile range. During pSWE measurements, the area of interest was identified 1–3 cm under the liver capsule in hepatic tissue, avoiding vessels and bile ducts, and placing the probe in the right intercostal region intersecting the right axillary axis, applying minimal pressure [10]. A valid TE test was considered with 10 valid measurements with an interquartile range (IQR)/median ≤30%; and a valid pSWE in case of 10 valid measurements with an IQR/median ≤30%. Both procedures were performed on the same day. Staging of fibrosis was assigned according to Metavir classification (F0, absent; F1, enlarged fibrotic portal tract; F2, periportal or initial portal–portal septa but intact architecture; F3, architectural distortion but no obvious cirrhosis; and F4, cirrhosis) [14]. Cutoffs were derived from the 43rd Annual Meeting of the European Association for the Study of the Liver (EASL) [15]. In particular, liver stiffness values ≥7.0 KPa, 8.7 KPa, 10.3 Kpa, and 14 kPa were considered as representative of mild (F1), moderate (F2), advanced fibrosis (F3) and cirrhosis, respectively. Patients were sub-classified according to the liver disease etiology [hepatitis B virus (HBV), alcoholic liver disease (ALD), hepatitis C virus (HCV), non-alcoholic steatohepatitis (NASH), autoimmune hepatitis (AIH), primary biliary cholangitis (PBC), primary sclerosing cholangitis (PSC), unknown hypertransaminasemia, and other causes)]. Demographic and clinical data were obtained for each patient.

The primary objective was to determine the accuracy and concordance of pSWE for the evaluation of liver fibrosis in chronic liver diseases of different etiologies compared to Fibroscan. The secondary objective was to evaluate whether the results agreement between the two methods was influenced by the etiology of liver disease or the severity of liver fibrosis.

Continuous data were expressed as median with interquartile range and were compared by Mann–Whitney U; discrete data were expressed as proportions and were compared by chi-squared test. The correlation between continuous variables was performed by calculating Spearman’s correlation coefficient r and then with linear regression. Bland–Altman plots were also drawn to study the agreement between data, calculating the bias value and its 95% confidence interval (95% CI). If the bias is within the confidence interval of the mean, the two methods may be used interchangeably. Receiver operating characteristics (ROC) curves were drawn to calculate sensitivity, specificity, and area under the curve (AUC). We planned to recruit 176 subjects as the minimum sample size, by estimating the sensitivity and specificity of 90% of the pSWE compared to the Fibroscan, with an accuracy of 10% and a confidence interval of 95%. All statistical analyses were performed two tailed, setting *p* < 0.05 as limit of significance. The statistical software GraphPad Prism version 5.0 was used.

## 3. Results

### 3.1. Patients’ Characteristics

One hundred and eighty-four patients were recruited (Table 1), 77 females and 107 males. Their mean age was 56.9 ± 12.5 years. The most common indication to elastography was NASH (*n* = 61), HBV (*n* = 29) and HCV (*n* = 21): Among the other etiologies, ALD was found in 7 subjects, a cholestatic disorder (either PBC or PBC) in 16, and AIH in 11. Eighteen subjects underwent elastography for undetermined hypertransaminasemia. The remaining 21 patients showed other causes such as drug-induced liver disease, cystic fibrosis, and hematological disorders (thalassemia, myeloproliferative syndromes).

### 3.2. Overall Correlation

Overall, the correlation of LS between Fibroscan and pSWE was substantial (r = 0.68, 95% CI 0.59–0.75) with a *p* < 0.001. Linear regression showed a coefficient b = 0.94 ± 0.02 (*p* < 0.001), with a goodness of fit r^2^ = 0.91 (Figure 1a). The Bland–Altman plot, represented in Figure 1b, found a bias of −0.10 (−5.2 to 5.0), with only 11 values exceeding the 95% CI.

### 3.3. Sub-Analysis According to Liver Stiffness

When considering only patients with a LS of < 6 kPa (*n* = 119), we found a low r = 0.27 (0.09–0.44, *p* = 0.002), as shown in Figure 2. The b coefficient was 0.39 ± 0.14, *p* = 0.006 with a goodness of fit r^2^ = 0.06. Bias was 0.37 (−2.5 to 3.2), with 7 values out of the 95% CI range. The ROC curve showed that a cutoff of 6.25 kPa for pSWE had a sensitivity of 72.3% and a specificity of 92.4% to exclude significant fibrosis, with an AUC of 0.88.

By considering only patients with a LSM of < 8 kPa (*n* = 146), the correlation between TE and SWE was confirmed to be quite low, with a r = 0.42 (0.27 to 0.55, *p* < 0.001). The b coefficient was 0.60 ± 0.11, *p* < 0.001, with a goodness of fit r^2^ = 0.18. ROC curve analysis showed that a cutoff of 9.3 kPa for pSWE had a sensitivity of 73.7% and a specificity of 98.6% to exclude significant fibrosis, with an AUC of 0.95.

Thirty-four patients had an intermediate LS, between 6 and 10 kPa. We found a low r = 0.19 (−0.17 to 0.50, *p* = 0.29), as reported in Figure 3. The b coefficient was 0.42 ± 0.43, *p* = 0.33 with a goodness of fit r^2^ = 0.03. Bias was −0.81 (−5.96 to 4.34), with 9 values out of the 95% CI range. pSWE values of 6.05 and 9.05 had, respectively, a sensitivity of 78.5% and 47.7% and a specificity of 89.1% and 99.2% to select patients within the TE range 6–10.

When considering only patients with a relevant fibrosis, i.e., LS > 10 kPa (*n* = 31), we found an excellent r = 0.79 (0.60–0.90, *p* < 0.001), as shown in Figure 4. The b coefficient was 0.97 ± 0.06, *p* < 0.001 with a goodness of fit r^2^ = 0.91. Bias was −1.14 (−10.4 to 8.12), with only one value exceeding the 95% CI range. The ROC curve showed that a cutoff of 12.15 kPa for pSWE had a sensitivity of 74.2% and a specificity of 99.3% to detect relevant fibrosis, with an AUC of 0.98.

### 3.4. Sub-Analysis According to Etiology

The highest correlation was observed for HCV with r = 0.91 (95% CI 0.77–0.96, *p* < 0.001) and ALD with r = 0.99 (95% CI 0.91–0.99, *p* < 0.001). A slightly lower correlation was observed for HBV (r = 0.67), NASH (r = 0.62), PBC/PSC (r = 0.72), and other causes of liver disease (r = 0.67). An insufficient correlation was recorded for AIH (r = 0.50) and hypertransaminasemia of unknown origin (r = 0.53). Further details, including the bias of Bland–Altman plots, are summarized in Table 2. Scatterplots of such sub-analyses are reported in Figure 5.

## 4. Discussion

Estimation of liver fibrosis is a crucial endpoint for the management of chronic liver disorders, as it is fundamental to stratify the patient for the risk of complications and to guide the treatment. However, liver biopsy is no longer routinely performed since it is an invasive procedure with possible adverse events. Surrogate methods to estimate liver fibrosis are gaining growing consent as they are often simple and non-invasive. Among them, TE is one of the most widespread. More recently, other techniques, based on acoustic radiation force impulse, have been implemented on ultrasound machines. pSWE is one of them. pSWE has some advantages over TE; for example, it allows direct visualization of the target tissue, thus avoiding vessels or the biliary tree, and may be performed on patients with ascites. The European Federation of Societies for Ultrasound in Medicine and Biology (EFSUMB) recommend TE, pSWE, and 2D-SWE as first-line techniques to grade liver fibrosis and rule cirrhosis in chronic hepatitis C, but any LSM changes during follow-up or after a successful treatment should not modify the patient management due to the scarcity of data for the new direct acting antivirals (DAAs). Similarly, the same elastographic techniques can be useful to rule out cirrhosis in chronic hepatitis B, non-alcoholic fatty liver disease, and alcoholic liver disease, prior to the exclusion of acute hepatitis [16]. According to the most recent Baveno consensus, LSM by TE can be used in the non-invasive diagnosis and monitoring of clinically significant portal hypertension in patients with compensated advanced chronic liver disease [17]. In a study conducted on more than 300 chronic liver disease patients, the two elastographic methods correlated significantly with liver biopsy (r = 0.79 for SWE and r = 0.70 for TE). The AUC for the diagnosis of significant fibrosis was 0.88 for SWE and 0.84 for TE, while, for cirrhosis, AUC was, respectively, 0.93 and 0.90 [18]. In another study, the AUC of pSWE was 0.84 for F > 3 and 0.94 for cirrhosis [19]. Finally, in another multicentric study, Joo et al. [20] found a correlation of r = 0.88 between TE and pSWE values.

For these reasons, the European Association for the Study of the Liver (EASL), in its guidelines on non-invasive testing for liver fibrosis assessment [21], has integrated SWE in the statements. In particular, EASL underlined that pSWE has equivalent performance for diagnosing significant fibrosis or cirrhosis compared to TE, while it is less sensitive for intermediate stages of fibrosis. This is in perfect agreement with our findings, as we estimated a good correlation between TE and pSWE for high fibrosis stages (r = 0.79), while for intermediate grades the correlation was insufficient (r = 0.19). Furthermore, the AUC was 0.98 for advanced fibrosis, while pSWE was less accurate (AUC = 0.88) when ruling out LS values of <6 kPa at TE. This could be explained by the so-called “spectrum bias”: if extreme grades of fibrosis (F0 and F4) are over-represented in a population, the sensitivity and specificity of a diagnostic method tend to be better than in a population with intermediate stages of fibrosis (F1 and F2). This could be a limitation even of our study, as we enrolled only 34 subjects with LS ranging from 6 to 10 kPa, while most of them had a LS of <6 kPa. The improved accuracy for high level of fibrosis detection is in agreement with the results of Valente et al. [22], who calculated a cut off of 11.6 kPa for pSWE, very close to ours (12.15 kPa). Another possible limitation is that we did not collect laboratory analyses assessing liver function in such patients. However, the aim of this study was simply to compare the two techniques, without correlating results with disease stages, nor evaluating fibrosis with blood tests such as platelet count. In this regard, we are planning a future study with this aim.

We noticed a correlation of 0.68 between TE and pSWE, which is slightly inferior to what was found by Foncea et al. (r = 0.88) [23]. However, the b index was very close to 1, thus underlining that pSWE provides numerical values that are directly proportional to TE in a 1:1 ratio. Finally, the Bland–Altman plots provided a bias close to zero, with only eleven values out of the 95% CI range, thus confirming a good concordance between the two techniques. Notably, most of the outliers (nine) were observed when restricting the analysis to the 6–10 kPa group, a result which confirms that the only Achilles heel for pSWE may be the intermediate fibrosis range. 

The sub-analysis according to the etiology of liver disease provided some interesting data. The best concordance was recorded for HCV, while it was less strong for HBV. This is in perfect agreement with EASL statements [21], which confirm that pSWE is better validated for HCV than HBV. NASH was the most common indication of liver stiffness measurement in our cohort. However, the concordance was not optimal (r = 0.62). Indeed, EASL stated that non-invasive tests are less accurate compared to liver biopsy for NASH in order to detect fibrosis, despite some recent encouraging results. Roccarina et al. [24] found a r = 0.80, while the AUC was higher for the detection of F4 (0.95), than for F1 (0.84): these results highlight that, even for NASH, low or intermediate grades of fibrosis are less accurately investigated by pSWE. AIH, PSC, and PBC did not show an excellent agreement. The validation of non-invasive testing for such a disease is indeed far from being found, as underlined also by EASL; therefore, more robust data and larger trials are necessary to assess fibrosis in such conditions. Another relevant message that may be drawn by our cohort is about unexplained hypertransaminasemia. These patients exhibited the worse correlation (r = 0.53); therefore, the LS measurement may not be optimal in this context, without a more precise diagnostic framework.

It should be acknowledged that other elastography techniques are available for the measurement of liver stiffness, which can be divided according to different parameters. Strain elastography (SE) and strain rate imaging (SRI) are quasi-static strain imaging techniques in which a displacement force is applied either mechanically from the outside, pushing with the ultrasound probe, or internally by muscle contractions or physiological cardiovascular and pulmonary pulsations. Multiple frame images of axial displacement are created and converted into qualitative strain images according to the tissue hardness. Alternatively, acoustic radiation force impulse (ARFI) imaging is a dynamic elastography method which qualitatively evaluates tissue displacement in a localized region of interest (ROI) through focused ultrasound-based repetitive pulses; it produces higher quality images compared to the previous techniques but applies only to a limited area. The tissue displacement induced by ARFI also generates shear waves, which propagate in the liver parenchyma with a measurable speed, which varies based on the elasticity of the tissue. As previously described, shear wave speed is measured in a defined ROI in pSWE, without the formation of colored elastographic maps; by applying the same ARFI technique sequentially in multiple locations, the ROI is widened and it creates multidimensional real-time quantitative images: this is called two dimensional SWE (2D-SWE). The elastograms produced can be visualized on a background of B-mode ultrasound so that off-target measurement (biliary tree, vessels, focal lesions) is avoided. All the described techniques have advantages and drawbacks, and they may not always be present in all the commercially available systems [16,25,26]. Magnetic resonance elastography (MRE) is another non-invasive technique found to be more accurate in measuring liver stiffness than TE when comparing the results with liver biopsy taken as a standard test [26]. Mechanical shear waves are produced by an external pneumatic device attached to the patients’ right abdomen which creates tissue displacement by low-frequency waves, detected by the MR and converted in elastograms; the operator can then select different ROIs or calculate a mean liver stiffness. Indeed, it has high intra- and inter-observer variability, leading to reliable results but it is limited by higher costs, patients’ cooperation, patient-related characteristics, and an incapability of distinguishing between the various etiologies of liver fibrosis [27].

## 5. Conclusions

In conclusion, our research highlighted some key points about pSWE. Despite a good agreement with TE being observed, there are relevant variations in the accuracy which depend on the etiology of liver disease and on the staging of fibrosis. Autoimmune liver diseases are the field that needs the most relevant implementation in study and research, as the correlation between the two techniques is less strong. Similarly, NASH needs to be explored more in depth. Other studies are encouraging, but the extremely broad spectrum of the disease could explain why, in our cohort, the accuracy was not optimal. On the other hand, pSWE could be an optimal replacement for TE, when this last exam is not available for viral hepatitis. Finally, similarly to TE, the measurement of liver stiffness by pSWE improves in precision as the fibrosis progresses, therefore it could be more useful to rule out cirrhosis in doubtful cases.

## Figures and Tables

**Figure 1 diagnostics-14-00604-f001:**
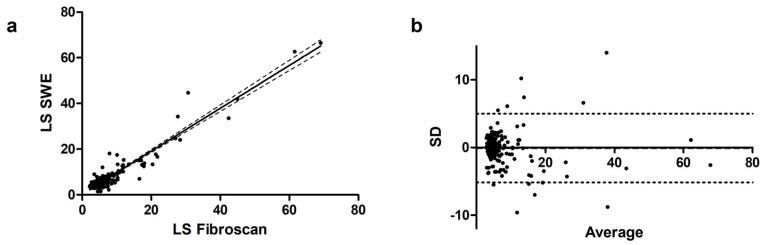
(**a**) Scatterplot and linear regression line of TE compared to pSWE in the overall cohort of patients; (**b**) Bland–Altman plot in the overall cohort of patients.

**Figure 2 diagnostics-14-00604-f002:**
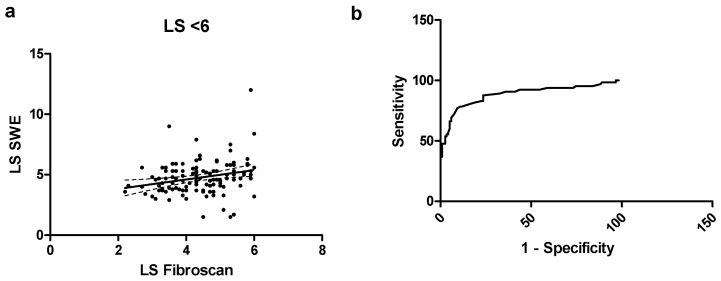
(**a**) Scatterplot for correlation for the subgroup of patients with LS < 6 kPa; (**b**) ROC curve for the subgroup of patients with LS < 6 kPa.

**Figure 3 diagnostics-14-00604-f003:**
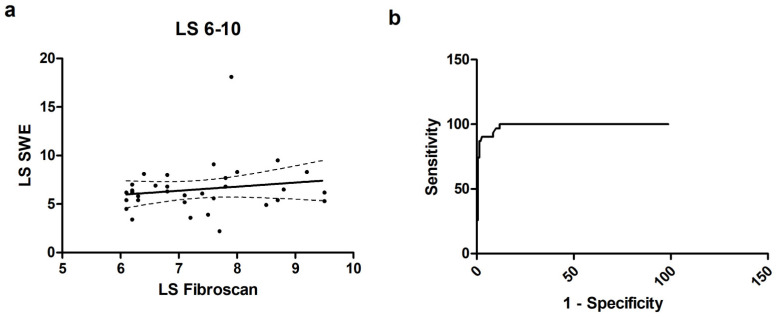
(**a**) Scatterplot for correlation for the subgroup of patients with LS 6–10 kPa; (**b**) ROC curve for the subgroup of patients with LS 6–10 kPa.

**Figure 4 diagnostics-14-00604-f004:**
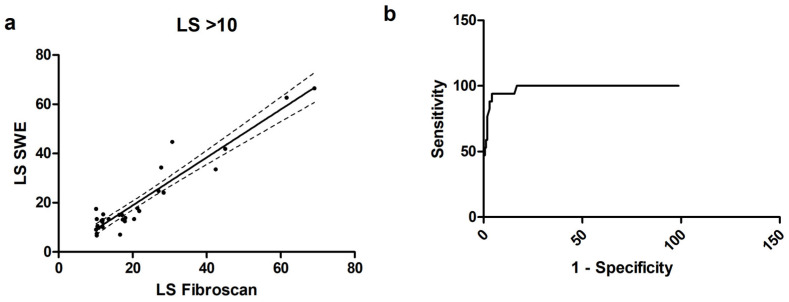
(**a**) Scatterplot for correlation for the subgroup of patients with LS > 10 kPa; (**b**) ROC curve for the subgroup of patients with LS > 10 kPa.

**Figure 5 diagnostics-14-00604-f005:**
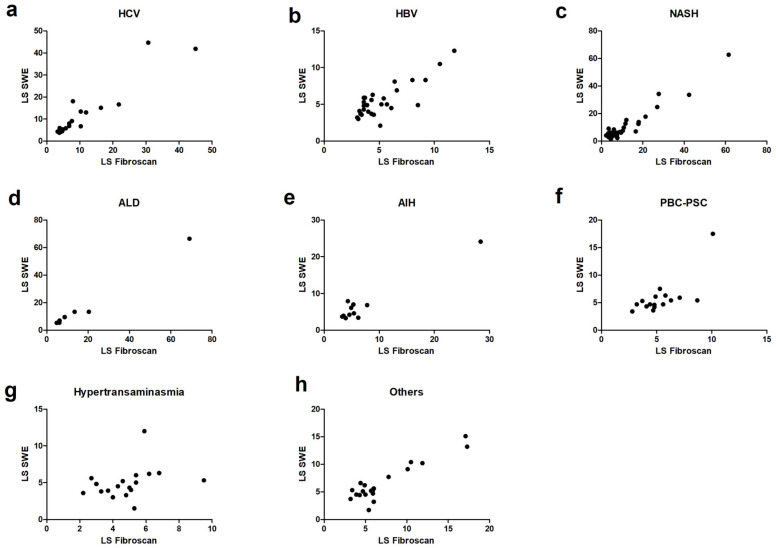
Scatterplots reporting the correlation analysis for subgroups according to the etiology: HCV (**a**), HBV (**b**), NASH (**c**), ALD (**d**), AIH (**e**), PBC-PSC (**f**), unexplained hypertransaminasemia (**g**), and other causes (**h**).

**Table 1 diagnostics-14-00604-t001:** Baseline features of enrolled population.

Population characteristics	Mean ± SD or *n* (%)
Age	56.9 ± 12.5
Male/female ratio	107/77
Liver disease etiology - HBV - HCV - NASH - PBC/PSC - AIH - ALD - Undetermined hypertransaminasemia - Other	29 (15.8%) 21 (11.4%) 61 (33.1%) 16 (8.7%) 11 (6.0%) 7 (3.8%) 18 (9.8%) 21 (11.4)

**Table 2 diagnostics-14-00604-t002:** Correlation parameters and bias according to the etiology of liver disease.

	Median Stiffness Measured by		Correlation			Bias
	TE	pSWE	r	95% CI	*p*	
HBV	4.3	4.9	0.67	0.39–0.84	<0.001	0.21
ALD	8.7	9.5	0.99	0.91–0.99	<0.001	−1.18
HCV	6.8	6.7	0.91	0.77–0.96	<0.001	1.12
NASH	5.2	5.6	0.62	0.43–0.76	<0.001	−0.57
PBC/PSC	4.85	5.0	0.72	0.31–0.89	0.002	0.45
AIH	4.9	4.6	0.50	0.10–0.78	0.11	−0.24
Hypertransaminasemia of unknown origin	4.9	4.65	0.53	0.06–0.80	0.02	0.06
Other causes	5.7	5.3	0.67	0.33–0.86	0.008	−0.53

HBV: hepatitis B virus; ALD: alcoholic disease; HCV: hepatitis C virus; NASH: non-alcoholic steatohepatitis; PBC: primary biliary cholangitis; PSC: primary sclerosing cholangitis; AIH: autoimmune hepatitis.

## Data Availability

Data is contained within the article.

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
