# Peer review of "Comparison of Transient Elastography and Point Shear Wave Elastography for Analysis of Liver Stiffness: A Prospective Study"

_diagnostics, 2024, doi:10.3390/diagnostics14060604_

Round 1

Reviewer 1 Report

Comments and Suggestions for Authors

Review of the manuscript: “Comparison of Transient Elastography and Point Shear Wave Elastography For Analysis of Liver Stiffness: a Prospective Study,” by Losurdo G. at al.

Major Comments:

The paper delivers intriguing insights into the correlation between liver elasticity measurements obtained via Fibroscan and SWE. Nevertheless, the methodology section, particularly the calculation of Receiver Operating Characteristic (ROC) curves, necessitates further clarification. It remains ambiguous whether the analysis was conducted based on Fibroscan data or actual disease diagnoses. The absence of ROC curve estimations for Fibroscan and its comparison with SWE results raises questions. Furthermore, the application of fibrosis staging within this analysis is not clearly articulated. Should the ROC analysis rely solely on Fibroscan outcomes, devoid of disease-specific data, its relevance could be deemed limited, given Fibroscan acknowledged diagnostic accuracy range (AUC 80-90%).

The manuscript mentions "point shear wave elastography" without clearly distinguishing it from conventional shear wave elastography. A detailed explanation highlighting the differences between these terms would enhance the reader's comprehension.

Minor Comments:

Page 2, Line 53: The statement regarding the differing physical properties utilized by each elastography method is vague. A clarification is needed, as all elastography techniques fundamentally measure tissue elasticity.

Materials and Methods Section: It's unclear whether Body Mass Index (BMI) measurements were included in the study parameters and what the average BMI among participants was.

Page, Line 105: The usage of the Metavir classification for fibrosis staging within the analysis is not explained.

Page 3, Line 108: The document should specify the cutoff values derived from the 43rd Annual Meeting of the European Association for the Study of the Liver (EASL).

Figure 2(a): The Y-axis requires a caption.

Page 4, Line 155: The paper does not clarify whether the ROC analysis for a cutoff of 6.25 kPa pertains to Fibroscan measurements.

Discussion on ROC Curve (Fig. 3b): The document lacks a discussion segment for the ROC curve depicted in Figure 3b.

Page 4, Line 164: The term "relevant fibrosis" used to describe the cutoff of 12.15 kPa for pSWE is ambiguous and needs a precise definition.

Table 1: Incorporating average or median elasticity modulus values for specific diseases for both methodologies would provide a clearer comparison.

Author Response

The paper delivers intriguing insights into the correlation between liver elasticity measurements obtained via Fibroscan and SWE. Nevertheless, the methodology section, particularly the calculation of Receiver Operating Characteristic (ROC) curves, necessitates further clarification. It remains ambiguous whether the analysis was conducted based on Fibroscan data or actual disease diagnoses. The absence of ROC curve estimations for Fibroscan and its comparison with SWE results raises questions. Furthermore, the application of fibrosis staging within this analysis is not clearly articulated. Should the ROC analysis rely solely on Fibroscan outcomes, devoid of disease-specific data, its relevance could be deemed limited, given Fibroscan acknowledged diagnostic accuracy range (AUC 80-90%).

We confirm that the ROC curves were attained considering the Fibroscan as the gold standard. We agree that this may be a relevant limitation and, for this reason, we have discussed this point in the appropriate section. However, we would like to point out that comparing pSWE by ROC curve with the original gold standard (liver biopsy) may not be easily attainable, as:

1) Liver biopsy is invasive and not devoid of possible severe complications; consequently, it cannot be offered to all patients, especially those without significant fibrosis, which were the majority in our series;

2) Fibroscan is considered itself as a gold standard and has replaced biopsy in most of cases of routine practice, as suggested by the EASL guidelines (reference no 21 of revised manuscript).

The manuscript mentions "point shear wave elastography" without clearly distinguishing it from conventional shear wave elastography. A detailed explanation highlighting the differences between these terms would enhance the reader's comprehension.

Point shear wave elastography refers to the fact that the measurement of stiffness is done by targeting a single point in the localized region of interest (ROI). This is different from 2D-SWE, the other shear wave elastography technique, in which the device calculates the mean stiffness within a larger area.

A detailed explanation was enclosed in revised manuscript.

Minor Comments:

Page 2, Line 53: The statement regarding the differing physical properties utilized by each elastography method is vague. A clarification is needed, as all elastography techniques fundamentally measure tissue elasticity.

Since all elastographic methods are based on the principle that tissue elasticity may be indirectly estimated by measuring the propagation of an ultrasound pulse (for SWE) or a mechanical stimulus (for TE), we agree with the referee that this sentence may be misleading, therefore we removed it. Nevertheless, the differences in the principles of both techniques have been described in the Introduction in the last three paragraphs.

Materials and Methods Section: It's unclear whether Body Mass Index (BMI) measurements were included in the study parameters and what the average BMI among participants was.

BMI were not considered for the analysis in our study.

Page, Line 105: The usage of the Metavir classification for fibrosis staging within the analysis is not explained.

The Metavir classification (F0-1-2-3-4) is the traditional way to measure fibrosis for Fibroscan, derived from the homologous histologic classification. Cut-offs were derived from the 43rd Annual Meeting of the European Association for the Study of the Liver (EASL), as better explained in the next answer (reference no 15 of revised manuscript).

Page 3, Line 108: The document should specify the cutoff values derived from the 43rd Annual Meeting of the European Association for the Study of the Liver (EASL).

Liver stiffness values ≥ 7.0 KPa, 8.7 KPa, 10.3 KPa and 14 kPa were considered as representative of mild (F1). moderate (F2), advanced fibrosis (F3) and cirrhosis, respectively, as reported in the text of revised manuscript.

Figure 2(a): The Y-axis requires a caption.

We corrected figure 2a.

Page 4, Line 155: The paper does not clarify whether the ROC analysis for a cutoff of 6.25 kPa pertains to Fibroscan measurements.

The cited cut off refers to the value measured by pSWE, considering TE as gold standard. The text was changed as suggested.

Discussion on ROC Curve (Fig. 3b): The document lacks a discussion segment for the ROC curve depicted in Figure 3b.

We thank the reviewer for pointing out this oversight. In the revised manuscript, we added that pSWE values of 6.05 and 9.05 had respectively a sensitivity of 78.5% and 47.7% and a specificity of 89.1% and 99.2% to select patients within the TE range 6-10.

Page 4, Line 164: The term "relevant fibrosis" used to describe the cutoff of 12.15 kPa for pSWE is ambiguous and needs a precise definition.

The term “relevant fibrosis” was derived from EASL guidelines, as it depicts patients with a fibroscan LS > 10 kPa. Herein, we calculated that pSWE has the best performance to detect relevant fibrosis when the cut-off was set at 12.15 kPa.

Table 1: Incorporating average or median elasticity modulus values for specific diseases for both methodologies would provide a clearer comparison.

In the revised table, we added median values per each etiology as measured by TE and pSWE.

Reviewer 2 Report

Comments and Suggestions for Authors

The use of the Fibroscan may be desirable, but it does not reflect current device developments. There should also be a critical comparison with ARFI and new kinds of 2D shear wave technolgies or at least discussed. Current EFSUMB guidleines are to be considered. What are the benefits of evalation with klPA compared to m/s ? What is the confinement of the examination conditions, something ascite? How does the stellel value compare to MRI or surgical findings? The biopsy does not have to be representative. What is the significance of CEUS Perfusion?

Comments on the Quality of English Language

Please give more details description about other actually technologies for liver tissue fibrosis evaluations like STE, 2D shear wave, ARFI etc and transducer techologies.

Author Response

The use of the Fibroscan may be desirable, but it does not reflect current device developments. There should also be a critical comparison with ARFI and new kinds of 2D shear wave technologies or at least discussed.

What are the benefits of evalation with klPA compared to m/s ?

There is no benefit of using kPa over m/s as unit of measurement. The device used for this study provided the measure in kPa by default. The two values can be converted into each other, as shown in doi: 10.7150/thno.18650.

What is the significance of CEUS Perfusion?

We did not discuss CEUS perfusion as it is very far from the goals of our study.

Please give more details description about other actually technologies for liver tissue fibrosis evaluations like STE, 2D shear wave, ARFI etc and transducer techologies.

Other elastography techniques are available for the measurement of liver stiffness, which can be divided according to different parameters. Strain elastography (SE) and strain rate imaging (SRI) are quasi-static strain imaging techniques in which a displacement force is applied either mechanically from the outside pushing with the ultrasound probe or internally by muscle contractions or physiological cardiovascular and pulmonary pulsations. Multiple frame images of axial displacement are created and converted into qualitative strain images according to the tissue hardness. Differently, acoustic radiation force impulse (ARFI) imaging is a dynamic elastography method which qualitatively evaluates tissue displacement in a localized region of interest (ROI) through focused ultrasound-based repetitive pulses; it produces higher quality images compared to the previous techniques even if applied only to a limited area.  The tissue displacement induced by ARFI also generates shear waves, which propagate into the liver parenchyma with a measurable speed, which varies based on the elasticity of the tissue. As previously described, shear wave speed is measured in a defined ROI in pSWE, without the formation of coloured elastographic maps; by applying the same ARFI technique sequentially in multiple locations, the ROI is widened and it creates multidimensional real-time quantitative images: this is called two dimensional SWE (2D-SWE). Produced elastograms can be visualized on a background of B-mode ultrasound so that off-target measurement (biliary tree, vessels, focal lesions) is avoided. All the described techniques have advantages and drawbacks, and they may not always be present in all the commercially available systems (16, 14, 25-27 references of revised manuscript).

Current EFSUMB guidleines are to be considered.

The European Federation of Societies for Ultrasound in Medicine and Biology (EFSUMB) recommend TE, pSWE and 2D-SWE as first-line techniques to grade liver fibrosis and rule cirrhosis in chronic hepatitis C, but any LSM changes during follow-up or after a successful treatment should not modify the patient management due to the scarcity of data for the new direct acting antivirals (DAAs) effects on this parameter. Similarly, the same elastographic techniques can be useful to rule out cirrhosis in chronic hepatitis B, non alcoholic fatty liver disease and alcoholic liver disease, prior to the exclusion of acute hepatitis. According to the most recent Baveno consensus, LSM by TE  can be used in the non-invasive diagnosis and monitoring of clinically significant portal hypertension in patients with compensated advanced chronic liver disease [16, 25].

What is the confinement of the examination conditions, something ascite?

Unlike TE, in 2D-SWE it is possible to select a ROI through the ultrasound B-mode before the stiffness measurement, which can, therefore, be evaluated even in patients with ascites or central obesity.

How does the stellel value compare to MRI or surgical findings? The biopsy does not have to be representative.

Magnetic resonance elastography (MRE) is another non-invasive technique found to be more accurate in measuring liver stiffness than TE, when the results are compared  with liver biopsy taken as gold standard. Mechanical shear waves are produced by an external pneumatic device attached to the patient right abdomen generating tissue displacement by low-frequency waves, which are detected by the MR and converted in elastograms. Thus, the operator can select different ROIs or calculate a mean liver stiffness. Despite the high intra- and inter-observer variability, it may lead to reliable results, but it is limited by high costs, patient cooperation, patient-related characteristics and inadequacy to distinguish various etiologies of liver fibrosis [26, 27].

  1. Dietrich CF, Bamber J, Berzigotti A, Bota S, Cantisani V, Castera L, Cosgrove D, Ferraioli G, Friedrich-Rust M, Gilja OH, Goertz RS, Karlas T, de Knegt R, de Ledinghen V, Piscaglia F, Procopet B, Saftoiu A, Sidhu PS, Sporea I, Thiele M. EFSUMB Guidelines and Recommendations on the Clinical Use of Liver Ultrasound Elastography, Update 2017 (Long Version). Ultraschall Med. 2017 Aug;38(4):e16-e47. English. doi: 10.1055/s-0043-103952. Epub 2017 Apr 13. Erratum in: Ultraschall Med. 2017 Aug;38(4):e48. PMID: 28407655.
  2. Bamber J, Cosgrove D, Dietrich CF, Fromageau J, Bojunga J, Calliada F, Cantisani V, Correas JM, D'Onofrio M, Drakonaki EE, Fink M, Friedrich-Rust M, Gilja OH, Havre RF, Jenssen C, Klauser AS, Ohlinger R, Saftoiu A, Schaefer F, Sporea I, Piscaglia F. EFSUMB guidelines and recommendations on the clinical use of ultrasound elastography. Part 1: Basic principles and technology. Ultraschall Med. 2013 Apr;34(2):169-84. doi: 10.1055/s-0033-1335205. Epub 2013 Apr 4. PMID: 23558397.
  3. de Franchis R, Bosch J, Garcia-Tsao G, Reiberger T, Ripoll C; Baveno VII Faculty. Baveno VII - Renewing consensus in portal hypertension. J Hepatol. 2022 Apr;76(4):959-974. doi: 10.1016/j.jhep.2021.12.022. Epub 2021 Dec 30. Erratum in: J Hepatol. 2022 Apr 14;: PMID: 35120736.
  4. Park CC, Nguyen P, Hernandez C, Bettencourt R, Ramirez K, Fortney L, Hooker J, Sy E, Savides MT, Alquiraish MH, Valasek MA, Rizo E, Richards L, Brenner D, Sirlin CB, Loomba R. Magnetic Resonance Elastography vs Transient Elastography in Detection of Fibrosis and Noninvasive Measurement of Steatosis in Patients With Biopsy-Proven Nonalcoholic Fatty wLiver Disease. 2017 Feb;152(3):598-607.e2. doi: 10.1053/j.gastro.2016.10.026. Epub 2016 Oct 27. PMID: 27911262; PMCID: PMC5285304.
  5. Ozturk A, Olson MC, Samir AE, Venkatesh SK. Liver fibrosis assessment: MR and US elastography. Abdom Radiol (NY). 2022 Sep;47(9):3037-3050. doi: 10.1007/s00261-021-03269-4. Epub 2021 Oct 23. PMID: 34687329; PMCID: PMC9033887.

Reviewer 3 Report

Comments and Suggestions for Authors

This interesting paper by Losurdo et al. showed that TE had a good agreement with pSWE, there were differences in the accuracy which depend on the etiology of liver disease and on the staging of fibrosis. However, this study also had some limitations such as no lab for liver function. Therefore, some major concerns are shown as following:

1. According to 2022 EASL clinical practice guidelines: non-invasive liver tests for evaluation of liver disease severity and prognosis (PMID: 36051951), the cutoffs to distinguish fibrosis stages by LSM<8, 8-10 and >10kPa. It would better use the cutoffs from the most updated guideline. Could you change the cutoffs to LSM<8, 8-10 and >10kPa in subgroup analysis according to liver stiffness.

2. It would be very interesting and important to compare AUROC, Sensitivity, Specificity of TE and pSWE in LSM<8, 8-10 and >10kPa. Please try to show these data.

3. Please add one Table to show the clinical characteristics of the overall population with 184 patients.

4. Could you also show the clinical characteristics in subgroup of LSM<8, 8-10 and >10kPa, respectively?

5. Please describe the aim of this study in more details in line 73-74.

Author Response

This interesting paper by Losurdo et al. showed that TE had a good agreement with pSWE, there were differences in the accuracy which depend on the etiology of liver disease and on the staging of fibrosis. However, this study also had some limitations such as no lab for liver function. Therefore, some major concerns are shown as following:

  1. According to 2022 EASL clinical practice guidelines: non-invasive liver tests for evaluation of liver disease severity and prognosis (PMID: 36051951), the cutoffs to distinguish fibrosis stages by LSM<8, 8-10 and >10kPa. It would better use the cutoffs from the most updated guideline. Could you change the cutoffs to LSM<8, 8-10 and >10kPa in subgroup analysis according to liver stiffness.

We would like to thank the reviewer for raising up this point. However, when using the proposed cut offs, only five patients would belong to the 8-10 kPa group, and this would make the power of the statistical analysis very dismal. Therefore, we decided not to change cut offs of the original, analysis, but only to make an additional analysis in patients with LSM<8, by calculating in such group correlation, AUROC, sensitivity and specificity. These additional data were added in the revised manuscript as a new paragraph.

The added paragraph reads as follows:

“By considering only patients with a LSM < 8 kPa (n = 146), the correlation between TE and SWE was confirmed to be quite low, with a r = 0.42 (0.27 to 0.55, p < 0.001). The b coefficient was 0.60 ± 0.11, p < 0.001, with a goodness of fit r2 = 0.18. ROC curve analysis showed that a cutoff of 9.3 kPa for pSWE had a sensitivity of 73.7% and a specificity of 98.6% to exclude significant fibrosis, with an AUC of 0.95.”

  1. It would be very interesting and important to compare AUROC, Sensitivity, Specificity of TE and pSWE in LSM<8, 8-10 and >10kPa. Please try to show these data.

See answer to point 1.

  1. Please add one Table to show the clinical characteristics of the overall population with 184 patients.

We added a new table (table 1 in the revised manuscript), as suggested.

  1. Could you also show the clinical characteristics in subgroup of LSM<8, 8-10 and >10kPa, respectively?

See answer to point 1.

  1. Please describe the aim of this study in more details in line 73-74.

Requested change has been performed.

Round 2

Reviewer 1 Report

Comments and Suggestions for Authors

Authors did satisfy reviewer's comments.

Comments on the Quality of English Language

Appropriate

Reviewer 3 Report

Comments and Suggestions for Authors

The authors satisfactorily addressed my comments.